# Ammonia and Particulate Matter Emissions at a Korean Commercial Pig Farm and Influencing Factors

**DOI:** 10.3390/ani13213347

**Published:** 2023-10-27

**Authors:** Lak-yeong Choi, Se-yeon Lee, Hanna Jeong, Jinseon Park, Se-woon Hong, Kyeong-Seok Kwon, Mijung Song

**Affiliations:** 1Department of Rural and Bio-Systems Engineering, Chonnam National University, Gwangju 61186, Republic of Korea; cly6847@jnu.ac.kr (L.-y.C.); seyeonn@jnu.ac.kr (S.-y.L.); 208769@jnu.ac.kr (H.J.); 2Education and Research Unit for Climate-Smart Reclaimed-Tideland Agriculture, Chonnam National University, Gwangju 61186, Republic of Korea; icarus381@jnu.ac.kr; 3AgriBio Institute of Climate Change Management, Chonnam National University, Gwangju 61186, Republic of Korea; 4Animal Environment Division, National Institute of Animal Science, Wanju 55365, Republic of Korea; kskwon0512@korea.kr; 5Department of Environment and Energy, Jeonbuk National University, Jeonju 54896, Republic of Korea; mijung.song@jbnu.ac.kr; 6Department of Earth and Environmental Sciences, Jeonbuk National University, Jeonju 54896, Republic of Korea

**Keywords:** emission factor, seasonal variation, correlation analysis, piglets, growing–finishing pigs

## Abstract

**Simple Summary:**

This study examined the levels of certain pollutants, primarily ammonia and particulate matter (PM), emitted by a commercial pig farm in Korea. Understanding these levels is vital for reducing the environmental impact of pig farming and ensuring a sustainable future for this industry. Interestingly, the study found that previous guidelines in Korea might have overestimated these pollution levels, although the current findings are consistent with global research. Seasonal variations also impacted pollution levels: more ammonia was released in spring and autumn, while other pollutants increased in summer. Factors such as the number and age of the pigs significantly influenced these emission levels, as did airflow and ventilation. In essence, continued research in this area will help to improve pig farming practices in Korea, ensuring a balance between successful farming and environmental preservation.

**Abstract:**

Quantifying emission factors of ammonia and particulate matter (PM) in livestock production systems is crucial for assessing and mitigating the environmental impact of animal production and for ensuring industry sustainability. This study aimed to determine emission factors of ammonia, total suspended particles (TSPs), PM_10_, and PM_2.5_ for piglets and growing–finishing pigs at a commercial pig farm in Korea. It also sought to identify factors influencing these emission factors. The research found that the emission factors measured were generally lower than those currently used in Korea, but were consistent with findings from individual research studies in the literature. Seasonal variations were observed, with ammonia emissions peaking in spring and autumn, and PM emissions rising in summer. Correlation analyses indicated that the number of animals and their average age correlated positively with both ammonia and PM emission factors. Ventilation rate was also positively correlated with PM emissions. Future extended field measurements across diverse pig farms will offer deeper insights into the emission factors of pig farms in Korea, guiding the development of sustainable livestock management practices.

## 1. Introduction

Livestock production is a significant contributor to global emissions of ammonia and particulate matter (PM), negatively affecting the environment and public health. Historically, the increase in global meat consumption has led to an increase in livestock farming activities, resulting in increased related emissions [1]. Emissions of ammonia from livestock manure management can give rise to fine PM, significantly impacting human health by causing respiratory and cardiovascular diseases [2]. Furthermore, PM can cause environmental problems such as reduced visibility and the acidification of soils and water bodies [3,4]. The livestock industry contributes over 70% of total ammonia emissions and over 80% of PM emissions from agricultural sources [5,6,7,8]. Given the rapid urbanisation trends, the proximity of farms to urban areas intensifies the need for emission controls to safeguard public health [9]. This situation has raised international concerns, prompting several mitigation strategies and policies from global environmental bodies [10]. Therefore, it is paramount to reduce ammonia and PM emissions from livestock for sustainable agriculture and better public health.

Measuring emission factors for ammonia and PM from livestock facilities is essential for understanding the impact of these emissions and for developing effective strategies to reduce them. Accurate measurements ensure compliance with regulatory standards and inform policy making for sustainable agricultural practices [11]. Emission factors represent the pollutants emitted per unit of animal production. Multiple methods are available to measure ammonia and PM emission factors from livestock facilities, including chamber-based methods [12,13], mass balance methods [14], micrometeorological methods [15], and inverse dispersion methods [16,17]. Each method has advantages and limitations, depending on factors such as the type of animal housing, investigation area size, and available equipment. Chamber-based methods excel in short-term emission rate measurements but might not capture a farm’s long-term emissions. In contrast, micrometeorological methods, ideal for large areas, demand specialised equipment and expertise [13,15].

A significant challenge in assessing ammonia and PM emission factors from livestock facilities is the need for long-term and continuous measurements to capture the temporal and spatial variations in emissions [13,18]. The emission factors can vary significantly based on factors such as temperature, humidity, animal conditions, and manure management practices [8,13,15,19,20]. As climate patterns shift and livestock rearing practices advance, predicting emissions becomes more complex [21]. Furthermore, most past studies have primarily focused on emission factors in research farms or small-scale pig facilities [22,23]. Few studies have examined modern commercial pig farms with advanced ventilation control systems. Emission factors, influenced by various factors, should be continuously monitored in conjunction with the development of pig farm facilities and the industry. Therefore, accurately measuring emission factors in modern pig farms requires long-term monitoring, which can be both logistically challenging and expensive.

This study aimed to measure the emission factors for ammonia and PM from a commercial pig farm in Korea and compare the results with literature values to improve the accuracy of emission estimates. The collected data were also analysed to understand the variability of emissions based on season and pig growth stage. Various factors influencing the emission factors were explored through a correlation study. This study aims to contribute to this important goal by providing new data on emission factors for ammonia and PM, while discussing the challenges and limitations of current measurements.

## 2. Materials and Methods

### 2.1. Experimental Setup

The study was conducted at a farrow-to-finish commercial pig farm situated in Jangseong City, southern Korea (latitude 35°21′2″ N, longitude 126°46′52″ E). The farm housed approximately 9000 pigs across seven newly built buildings equipped with modern facilities. The farm’s dimensions and layout conformed to the standard configurations for pig farms in Korea, as referenced from the Ministry of Agriculture, Food and Rural Affairs [24]. All pig production units had slotted floors, and a liquid manure pit recirculation system managed the manure in the pits under the slotted floors. This system aimed to decrease odour emission through biological treatment processes. Ammonia and PM measurements were taken from piglet pens (4–9 weeks of age) and growing–finishing pig pens (18–26 weeks of age) over a period of one and a half years starting in May 2020.

#### 2.1.1. Pig Buildings

Experiments were separately conducted for piglets and growing–finishing pigs. The piglet house consisted of a central corridor and eight rooms along the corridor. For measurements, the second room from the entrance was chosen. This room measured 11.8 × 8.7 × 2.6 m (L × W × H) and contained a central passage with eight pens on both sides. Each pen could house up to 50 piglets, making the total capacity of the room 400 piglets (Figure 1). The growing–finishing pigs were kept in three structurally identical buildings, and one of the three buildings was selected for every measurement. Each building had dimensions of 75 × 13 × 4.7 m (L × W × H), featured passages and 28 pens, and could accommodate up to 1100 pigs (Figure 2).

#### 2.1.2. Ventilation Systems

The experimental buildings incorporated mechanical ventilation systems. In the piglet room, fresh air was supplied from the corridor through openings on the front wall between the corridor and the room, as well as on the ceiling. Four small fans were mounted on the back wall to extract air, one of which was operated continuously to ensure minimum ventilation while the remaining three were activated based on the ventilation needs. These fans were controlled automatically based on air temperature setpoints. However, the fan control system was outdated and lacked digital monitoring; hence, fan operations were logged manually.

The growing–finishing pig building featured an automatic climate controller (DOL 634 CT, SKOV A/S, Glyngøre, Denmark) offering a combination of side and tunnel ventilation modes. In high external temperatures, the tunnel mode operated with 11 large exhaust fans on the end wall. During cooler temperatures, the side mode activated with 7 smaller exhaust fans installed on the roof. In both configurations, air entered through openings located on both side walls. The controller adjusted the number of operating fans to achieve desired ventilation rates and logged these data.

### 2.2. Emission and Ventilation Rate Measurements

Experiments were conducted 39 times (20 times for piglets and 19 times for growing–finishing pigs) from May 2020 to October 2021. While 4–8 experiments were performed each season, only 1 took place during winter due to restricted farm access for disease management. Each experiment involved setting up measurement instruments in the morning and retrieving them in the evening, maintaining at least a 6 h continuous measurement period.

For calculating emissions from both the piglet room and the growing–finishing pig building, ventilation rates along with concentrations of ammonia and PM at the air outlets were measured. The growing–finishing pig building’s ventilation rate was logged by its automatic climate controller during the experiments. However, the piglet room’s ventilation rate was measured manually. The actual ventilation rate of the four exhaust fans in the piglet room was measured every season using a flow hood (Model Testo 420, Testo, Inc., Lenzkirch, Germany). A video camera was used to record the operation of the exhaust fans during every experiment. This footage was analysed to calculate the operation time of each fan and determine the total ventilation rate of the piglet room during the experiment.

Ammonia and PM concentrations at the air outlet were detected using a portable gas detector (Gastiger 2000, Shenzhen Wandi Technology Co., Ltd., Shenzhen, China) and personal impactors (PEM, SKC Inc., Eighty-Four, PA, USA) connected to an Aircheck sampling pump (220-5000TC, SKC Inc., USA). PM_10_ and PM_2.5_ were collected on a PTFE filter (37 mm, 2.0 μm pore, SKC Inc., USA) at a flow rate of 4 L per minute. Total suspended particles (TSPs) were measured concurrently using a PV filter (37 mm, 5.0 μm pore, SKC Inc., USA) in a filter cassette (37 mm, SKC Inc., USA) without a personal impactor, operating at 2 L per minute. The HOBO probe (HOBO MX2301A, Onset Computer Corporation, Bourne, MA, USA) recorded air temperature and humidity during these measurements.

Each measurement set included equipment for ammonia and PM measurements, a HOBO probe, a protective box, and a tripod for accurate sampler positioning. To enhance measurement reliability and mitigate data loss due to equipment malfunctions, two sets were placed at each location. For air outlet measurements, sets were located on the exhaust fans’ inner side to prevent errors from turbulent airflows. Outdoor conditions like temperature, humidity, and ammonia and PM background concentrations were simultaneously gauged using sets placed in an open hill area within the farm.

In each experiment, the breeding information was provided by a farmer. The number of animals, average age of pigs, and marketed pigs per sow per year (MSY) were collected and used to analyse the correlation with the emission rate.

### 2.3. Calculation of Emission Factors

Ammonia concentration was measured in ppm and converted into mass concentration, mg m^−3^, using Equation (1):(1)$$C_{A}={\mathrm{ppm}}_{A}\times \frac{M_{A}}{V_{m}}\times \frac{273.15\;K}{T}\times \frac{P}{101.325\;\mathrm{kPa}},$$
where $C_{A}$ is the mass concentration of ammonia (mg m^−3^), ${\mathrm{ppm}}_{A}$ is the ammonia concentration in ppm, $M_{A}$ is the molecular weight of ammonia (17.031 g mol^−1^), $V_{m}$ is the volume of 1 mol at 1 atmospheric pressure at 0 °C (22.414 L mol^−1^), T is the measured air temperature (K), and P is the atmospheric barometric pressure (100.4436 kPa based on the site elevation of 82 m).

The concentration of PM was determined by dividing the mass of collected PM by the total volume of air that passed through the filters. The filters were weighed three times before and after measurement using a precise balance with a resolution of 1 μg (BM-22, A&D Company Ltd., Tokyo, Japan). The balance was kept in a closed chamber that contained a desiccator for drying filters and automatic dehumidifier units for maintaining a relative humidity of 30 ± 5%. All filters were dried in the desiccator for more than 24 h before weighing. In each experiment, field blank filters were prepared and treated during all measurement stages in the same manner as measurement filters except loading PM. The field blank filters were analysed to eliminate any uncertain errors that could be caused by filter contamination during all stages of handling and analysis [25]. The concentration of PM was calculated using Equation (2):(2)$$C_{PM}=\frac{\left(W_{S_{a}}-W_{S_{b}}\right)-\left(W_{B_{a}}-W_{B_{b}}\right)}{V_{s}\times \Delta t},$$
where $C_{PM}$ is the concentration of PM (μg m^−3^), W is the weight of the filters (μg), subscript letters S and B indicate the sampling filters and field blank filters, respectively, subscript letters a and b indicate weighing after and before the measurement, respectively, $V_{s}$ is the air sampling flow rate (m^3^ s^−1^), and Δt is the sampling time (s).

The emission rates of ammonia and PM during each experiment were calculated as follows [26]:(3)$$E_{i}=V_{ex_{i}}\times \left(C_{ex_{i}}-C_{bg_{i}}\right),$$

$E_{i}$ is the emission rate of each experiment (mg h^−1^), $V_{ex_{i}}$ is the average ventilation rate induced by exhaust fans (m^3^ h^−1^), and $C_{ex_{i}}$ and $C_{bg_{i}}$ are the average concentrations measured at the air outlets and background, respectively (mg m^−3^). Ideally, $C_{bg_{i}}$ should be measured at the air inlets to quantify the concentration of ammonia and PM entering the facility. However, due to limited measurement sets, it was assumed that the background concentration measured outdoors was equivalent to the concentration entering through the air inlets. Here, $C_{ex_{i}}-C_{bg_{i}}$ is defined as the emission concentration.

The emission rates of every experiment were averaged and divided by the number of animals to calculate the yearly emission factor using Equation (4):(4)$$\mathrm{EF}=\frac{\sum_{i}E_{i}}{n_{data}}\times \frac{0.00876}{n_{animals}},$$
where EF is the emission factor (kg animal^−1^ yr^−1^), $n_{data}$ is the number of emission measurements during a given period, $n_{animals}$ is the average number of animals during a given period, and 0.00876 is a constant for unit conversion.

### 2.4. Statistical Analysis

Emission factors for piglets and growing–finishing pigs were determined by averaging across the entire experimental period and then compared to values from the literature. Seasonal variations in emission factors were also analysed to investigate the seasonal effects on emissions. In addition, correlation analyses were conducted to identify the effects of various factors, including indoor climate (air temperature and relative humidity), outdoor conditions (air temperature, relative humidity, and background concentrations), operational information (ventilation rate and concentrations at air outlets), and breeding information (number and age of animals and MSY) on the emission factors. It is noteworthy that the MSY was evaluated monthly due to potential changes in pig production over time, making it more suitable to be compared with the monthly emission factors.

## 3. Results and Discussion

### 3.1. Emission Factors

All measurement data can be found in the Appendix A. Table 1 displays the calculated emission factors for piglets and growing–finishing pigs. The emission factors for ammonia were 0.31 and 1.85 kg animal^−1^ yr^−1^ for piglets and finishing pigs, respectively. In general, ammonia emission in confined pig buildings arises from under-floor manure storage [13]. As the pigs grew, their feed and manure production increased, leading to more ammonia emissions. It was evident that growing–finishing pigs produced higher amounts of TSP and PM compared to piglets. PM often originate from animal-related sources such as feed, faeces, and skin particles [8]. As pigs progressed through their growth stages, the dust from these sources correspondingly increased, causing the emission factors for TSP and PM to increase.

This study’s ammonia emission factors were benchmarked against existing literature. The Clean Air Policy Support System (CAPSS) of Korea suggested emission factors of 4.4 and 11.4 kg animal^−1^ yr^−1^ for piglets and growing–finishing pigs, respectively, based on experimental studies conducted in domestic pig farms [27]. The suggested values included emissions from buildings, compositing facilities, manure treatment facilities, and soil fertilisation. As indicated in Table 1, the emission factors from pig buildings were 1.9 and 5.2 kg animal^−1^ yr^−1^. The emission factor for TSP provided by CAPSS was based on US EPA speciate 4.0 data (2006). Meanwhile, the emission factors for PM_10_ and PM_2.5_ were sourced from the European inventory EMPT/CORINAIR [28]. Comparing our experimental results with those provided by CAPSS, we observed that the CAPSS emission factors were overestimated for all parameters, except for PM_2.5_ in growing–finishing pigs. This discrepancy could stem from CAPSS relying on foreign data. Given the distinct environmental and livestock conditions in Korea versus Europe and the US, differences are expected. Factors like breeding density, pit type, climate, and ventilation can influence emissions, but pinning down their combined effects is not straightforward, as demonstrated by the wide range of emission factors shown in Table 1. Additionally, the CORINAIR data used by CAPSS were outdated, and recent data [29] show a decrease in all emission factors. In Korea, advances in livestock facilities have improved breeding environments, which could potentially reduce ammonia and PM.

The emission factors estimated in this study were generally lower than those proposed in Korea and Europe. Specifically, there were significant differences in ammonia and TSP emission factors, while the PM_10_ emission factors were similar to those reported in Europe. The measured PM_2.5_ values in our study were higher than the European emission factors. However, it is worth noting that some individual studies listed in Table 1 have also reported similar emission factors to ours. This suggests that emission amounts can significantly vary across swine farms and underscores the importance of long-term measurements across diverse farms to determine representative values.

Notably, the emission factors of this study were derived from daytime measurements. The VERA test protocol suggests segmenting measurements into day and night if daily emission variations exist. However, due to restricted access and limited equipment, night-time measurements were not feasible in our study. The literature suggests that daytime ammonia emission factors are roughly 10% higher for piglets and 7% higher for fattening pigs than night-time values [30]. Although there are no studies comparing daytime and night-time emission factors for PM, daytime emissions are expected to be higher given their correlation to animal activity [22].

**Table 1 animals-13-03347-t001:** Emission factors were measured in this study (mean ± standard deviation) and obtained from the literature (kg animal^−1^ yr^−1^).

Literature	Piglets				Growing–Finishing Pigs			
	Ammonia	TSP	PM_10_	PM_2.5_	Ammonia	TSP	PM_10_	PM_2.5_
This study	0.31 ± 0.21	0.04 ± 0.03	0.03 ± 0.03	0.01 ± 0.01	1.85 ± 1.26	0.22 ± 0.19	0.16 ± 0.17	0.09 ± 0.10
Korean CAPSS [27]	1.9	0.54	0.18	0.029	5.2	1.26	0.42	0.069
EMPT/EEA [29]		0.27	0.05	0.002	3.7 ^1^, 2.8 ^2^	1.05	0.14	0.01
Literature review [31]			0.56–0.73 *				0.75–1.46 *	0.008
Literature review [23]					1.6–4.8		0.30–3.47 *	
Belgian pig farms [23]					2.20		0.10	0.008
Irish pig units [19]	0.40–1.09				2.52–4.34			
Experimental units [26]					2.38–2.53			
Chinese pig farms [32,33]	0.038–0.118					0.39	0.18	0.044

^1^ Housing, storage, and yards; ^2^ Manure application; * Unit: kg LU^−1^ yr^−1^.

### 3.2. Seasonal Variation in Emission Factors

The emission calculations for PM and ammonia were a product of the facility’s ventilation rate and the emission concentration. These emissions were highly influenced by climate and seasonal variations [8,20,32,34,35,36,37]. In mechanically ventilated buildings, high ventilation rates were maintained to alleviate the heat stress of animals in summer, which in turn reduced the concentration of indoor concentrations of PM and ammonia. Conversely, during winter, minimised ventilation rates were maintained to conserve thermal energy, resulting in elevated indoor concentrations. Although our current emission factors were calculated assuming a year-long uniformity, understanding the seasonal fluctuations in emissions is crucial for effective livestock management and local environmental conservation. In this study, the data collected during the entire experimental period were divided into four seasons: spring (March–May), summer (June–August), autumn (September–November), and winter (December–February). The emission factors were then calculated for each season.

The seasonal emission factors for ammonia, as delineated in Table 2, were as follows: 0.381, 0.299, 0.288, and 0.117 kg animal^−1^ yr^−1^ for piglets and 2.974, 1.017, 2.203, and 1.978 kg animal^−1^ yr^−1^ for growing–finishing pigs, corresponding to the spring, summer, autumn, and winter seasons, respectively. Interestingly, ammonia emissions were higher during the transitional seasons of spring and autumn, with a larger range of fluctuation, which could be attributed to significant changes in ventilation rate due to large daily temperature variations during the seasonal transition (Figure 3). Conversely, during the summer, the change in emission factors was relatively small, as daily temperature changes were less pronounced, with ventilation fans operating at their maximum or minimum ventilation capacities throughout the season. This pattern is expected to reflect similarly during the winter months. A comparison between summer and winter emissions revealed that summer ammonia emissions in piglet pens were approximately three times that of winter emissions, corroborating with the findings of Feng et al. [33]. However, the observed emission factors for growing–finishing pigs were unusual, as they were higher in winter than in summer. While ammonia generation typically increases at higher temperatures, resulting in increased emissions during summer [38], our study found that the facility’s indoor temperature was consistently maintained at 26 °C or above throughout the year. This could have offset the seasonal temperature effect on ammonia emissions. The facility’s liquid manure pit recirculation system might also have played a role. This system separated the solid fraction of manure for collection and recirculated the liquid fraction through aerobic treatment, which effectively decreased nitrogen levels, potentially reducing ammonia gas generation compared to conventional swine housing methods [39]. Another possible reason is the gas-to-particle conversion of ammonia gas to ammonium. This conversion is more prevalent in high-humidity environments [40,41,42]. Given that approximately 60% of Korea’s annual rainfall occurs during the summer, leading to heightened humidity, it is plausible that extended manure storage in growing–finishing pig facilities during these conditions might encourage this ammonia-gas-to-ammonium conversion. This could explain the lower-than-anticipated ammonia emissions in summer. Moreover, the significantly increased PM_2.5_ emissions observed in summer could align with this hypothesis.

This study also evaluated the seasonal emission factors of TSP, PM_10_, and PM_2.5_ in both piglet and finishing pig pens. For piglets, the seasonal TSP emission factors across spring, summer, fall, and winter were 0.044, 0.048, 0.039, and 0.009 kg animal^−1^ yr^−1^, respectively. For finishing pigs, the values were 0.249, 0.282, 0.137, and 0.247 kg animal^−1^ yr^−1^ in the same order. PM_10_ emissions or piglets were 0.029, 0.048, 0.018, and 0.006 kg animal^−1^ yr^−1^ while, for finishing pigs, they were 0.133, 0.214, 0.077, and 0.154 kg animal^−1^ yr^−1^ in the same order. PM_2.5_ emissions for piglets were 0.011, 0.022, 0.007, and 0.002 kg animal^−1^ yr^−1^ and, for growing–finishing pigs, they were 0.054, 0.193, 0.045, and 0.052 kg animal^−1^ yr^−1^. In contrast to ammonia emissions, PM emission factors peaked during summer, which was in agreement with the findings of Van Ransbeeck’s research [23]. The variability was also most pronounced in summer. While winter emissions were lower in piglet pens, the fattening pig pens exhibited emissions similar to or even higher than the transitional seasons, showcasing a distinct pattern compared to ammonia emission factors.

The study determined that seasonal variations in PM and ammonia emissions were relatively small in the piglet facilities. In contrast, these variations were more pronounced in the growing–finishing pig facilities. Notably, the indoor concentrations in the growing–finishing pig facilities, skyrocketed during winter, primarily due to the reduced ventilation. This led to a significant increase in emission concentrations—about three to six times higher than in the autumn season. Therefore, even with the reduced ventilation rates, the emission factors remained elevated during winter.

### 3.3. Correlation Analysis

The ventilation rate and emission concentration are pivotal in calculating emission factors. These metrics are influenced by various factors, including external meteorological conditions, indoor climatic conditions, animal growth conditions, and farm operations and management. In this study, Pearson’s correlation coefficient was used to assess the interrelationship between these variables, as detailed in Table 3, Table 4, Table 5 and Table 6.

From the correlation analysis, a clear positive correlation was observed between the number of animals, their average age, and the ventilation rate. As pigs matured and gained weight, their breeding density also increased. This subsequently led to an increase in the total ventilation rate within the facility, aiming to maintain the required ventilation rate per pig, aligning with findings from prior research [8,43].

A pronounced positive correlation was identified between the ventilation rate and air temperatures both inside and outside the barn, with coefficients ranging from 0.488 to 0.498. This could be attributed to the consistent internal temperatures maintained in both piglet and finishing pig facilities, achieved by modulating the ventilation in relation to the external temperature. Moreover, there was a weak negative correlation between ammonia emission concentration and ventilation rate (−0.340). A more distinct negative correlation was noted between ammonia emission concentration and the external air temperature (−0.656) and inside the barn (−0.674), consistent with the findings of Van Ransbeeck et al. [23].

The emission factor of ammonia demonstrated a strong positive correlation with both the number of animals (0.679) and their average age (0.613). As pigs age, the quantity of faeces they produce increases. Consequently, with an increasing number of animals, the total amount of faeces generated in the pit also increases, resulting in a proportional increase in the amount of ammonia generated. The emission factor of ammonia exhibited a strong positive correlation (0.574) with emission concentration but a very weak positive correlation (0.222 with a *p*-value of 0.174) with ventilation, indicating a stronger association with emission concentration compared to ventilation, both of which were factors involved in the calculation of the emission factor. Notably, the ammonia emission factor demonstrated a significant negative correlation (−0.514) with indoor relative humidity. Philippe’s review paper corroborated these observations, suggesting that while indoor relative humidity might not directly influence ammonia emissions, factors like indoor temperature or ventilation might impact it. This, in turn, might introduce secondary effects driven by various complex factors, leading to a correlation with ammonia emission [38]. Supporting this study, our study revealed that the ammonia emission factor in summer is significantly lower than in other seasons. Considering Korea’s climate, summers tend to have high relative humidity, while relative humidity drops during the winter and transition seasons. Thus, it is plausible to deduce that reduced humidity correlates with increased ammonia emissions, with the seasonal differences further solidifying the relationship between ammonia emissions and relative humidity.

Regarding the PM emission factors, TSP, PM_10_, and PM_2.5_ showed a positive correlation with the number of animals and their average age. This is because PM in pig houses often originates from various sources, including animal activities, faeces, feed dust, and hair. These data suggest that as the size and age of the pig increase, the likelihood of fine dust emission also increases.

Among the factors affecting the TSP emission factor, the average age had the highest correlation coefficient (0.581), followed by the number of animals (0.547), relative humidity inside the barn (−0.529), and ventilation rate (0.476). For PM_10_, the order of strong correlation coefficients was ventilation rate (0.638), number of animals (0.488), and average age (0.473). For PM_2.5_, it was ventilation rate (0.735), average age (0.526), and number of animals (0.465). The PM emission factors were influenced by both indoor relative humidity and indoor air temperature. However, TSP emission factors were more significantly influenced by indoor relative humidity. This is attributed to the fact that when humidity levels are high, dust absorbs more moisture, causing it to aggregate and settle on surfaces such as floors and equipment [44]. Conversely, the emission factors of smaller particles like PM_10_ and PM_2.5_ were similarly affected by both indoor air temperature and relative humidity.

Compared to ammonia, the ventilation rate had a greater impact on PM emission factors than on emission concentrations. The correlation coefficients between ventilation rate and emission factors were 0.476 for TSP, 0.638 for PM_10_, and 0.735 for PM_2.5_. Notably, as particle size decreased, the positive correlation coefficient between ventilation rate and emission factor gradually increased. This happens because fine particles are more influenced by airflow and can move more freely along the streamline formed by the exhaust fan in the facility.

A prior study by Aarnink and Ellen [43] reported that dust generated inside livestock facilities can either remain airborne or adhere to facility interiors and animals. Airborne dust can also re-deposit. The movement of animals mainly causes this airborne dust. Dust that becomes dislodged and airborne is then affected by the airflow in the facility formed by ventilation. Ventilation primarily affects the sedimentation rate of dust, with larger particles settling faster and smaller particles being more easily transported by air currents. Therefore, there is a stronger correlation between the ventilation rate and the generation of smaller PM.

Besides the emission factor, the emission concentration of PM was significantly affected by temperatures. The outdoor temperature had correlation coefficients of −0.635, −0.687, and −0.468 for TSP, PM_10_, and PM_2.5_, respectively, greatly affecting emission concentration. Indoor temperature also showed significant correlations, with coefficients of −0.539, −0.503, and −0.411, respectively. Both outdoor and indoor temperatures majorly impact the ventilation operation in pig farms. As temperatures increase, ventilation frequency and volume also increase, leading to reduced emission concentrations. However, no significant relationship was observed directly between emission concentration and ventilation rate. Meanwhile, as shown in Table 3, with the ventilation rate showing positive correlations of 0.488 and 0.498 with the outdoor and indoor temperatures, respectively, it can be inferred that increasing temperatures leads to an increase in ventilation rates, which indirectly reduces emission concentrations.

Table 7 provides a comparative analysis of the monthly MSY values from May 2020 to October 2021, as reported by the farmer, against the average monthly emission concentrations and emission factors during the same period. The analysis revealed a distinct positive correlation between MSY and emission concentrations for several emission substances. For instance, there was a notable correlation between MSY and the concentrations of TSP and PM_10_ for piglets and ammonia for growing–finishing pigs. However, drawing a direct connection between the emission concentration (which can be viewed as the indoor concentration) and MSY proved elusive. A higher MSY suggests a greater number of pigs ready for the market. Given that the number of pigs on the farm directly correlates with emission concentration, this seems to explain the observed relationship between MSY and emission concentration.

When MSY was compared with the emission factor, it was challenging to establish a clear correlation between ammonia and PM. It was thus concluded that the MSY, which represents the productivity of the farm, and the emission factors of ammonia and PM were not statistically significant. Prior research also supports this conclusion, suggesting that MSY is more influenced by qualitative factors such as the impact of diseases, education levels of farm owners, worker proficiency, and regular farm consultation rather than by quantitative factors such as the concentrations of ammonia or PM in farms [45,46].

## 4. Conclusions

In this study, we conducted field experiments at a commercial pig farm over a period of one and a half years to determine the emission factors of ammonia, TSP, PM_10_, and PM_2.5_ for piglets and growing–finishing pigs. Although the CAPSS emission factors in Korea generally seemed overestimated, our findings were consistent with certain individual studies, highlighting the variability in emissions across different swine farms. Significant seasonal variations were observed, with transitional seasons such as spring and autumn showing an increase in ammonia emissions due to fluctuations in ventilation rates. In contrast, during summer and winter, the variation in emission factors was relatively small. Our correlation analysis further revealed the number of animals and their average age as major contributors to ammonia and PM emissions. Additionally, while ventilation rates positively influenced PM emissions, their correlation with ammonia was comparatively weaker. However, this study has certain limitations that need to be addressed. Firstly, it focused on a single farm, which may not reflect the situation across all Korean pig farms. Secondly, the study did not examine how different feeding practices and manure management strategies impact emissions. Thirdly, the study did not entirely adhere to the VERA test protocol. More comprehensive results might be achieved with 24 h measurements that include night-time.

Future research should address these limitations and further investigating the factors affecting emissions from pig farms in Korea. Established practices, like enhanced manure management and optimised feed strategies, which have shown promise in various studies, need to be investigated for their suitability and efficiency in the Korean context. Furthermore, it is also crucial to investigate the impact of emissions from pig farms on local air quality and human health. The feasibility of emission trading schemes and other economic incentives, such as subsidies or grants for adopting pollution control technologies or practices, should also be explored to promote sustainable farming practices. In conclusion, this study provides significant insights into emissions from Korean pig farms and can inform the development of more sustainable and environmentally friendly livestock management practices.

## Figures and Tables

**Figure 1 animals-13-03347-f001:**
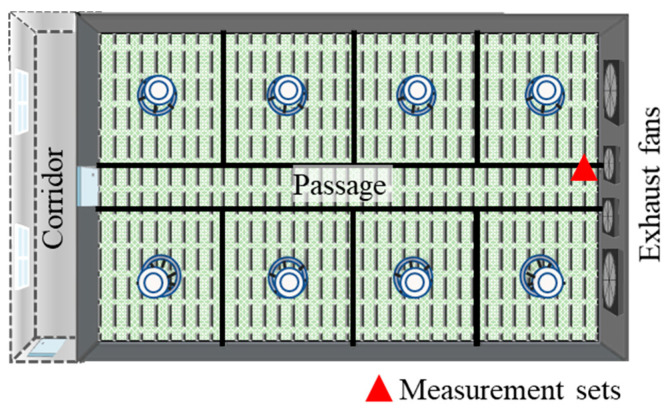
Schematic layout of piglet room selected for measurement.

**Figure 2 animals-13-03347-f002:**
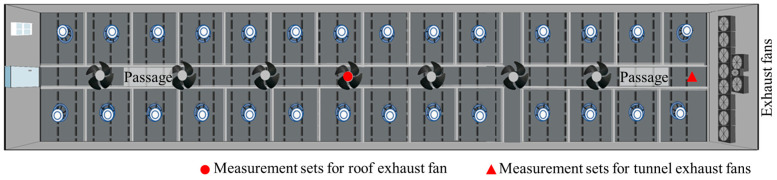
Schematic layout of the selected growing–finishing pig building for measurement.

**Figure 3 animals-13-03347-f003:**
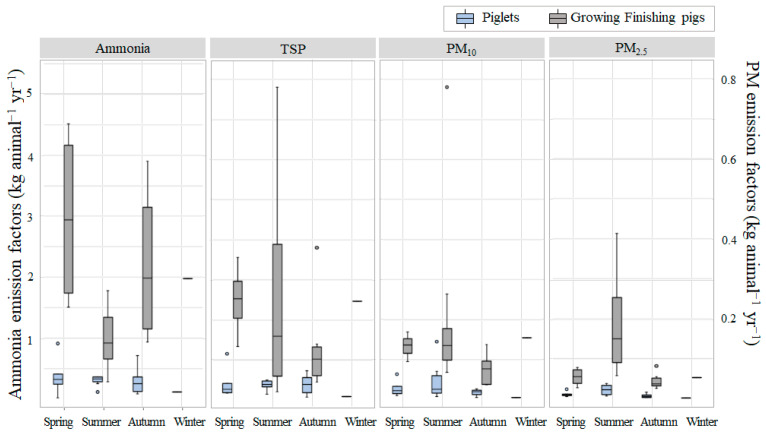
Seasonal variations in emission factors in piglet and growing–finishing pig facilities.

**Table 2 animals-13-03347-t002:** Seasonal variation in emission concentration, ventilation rate, and emission factor (mean ± standard deviation).

Data	Piglets				Growing–Finishing Pigs			
	Spring	Summer	Autumn	Winter	Spring	Summer	Autumn	Winter
VR ^1^	5069 ± 5078	17,613 ± 3816	3706 ± 1889	934 ± 0	53,511 ± 25,224	213,068 ± 55,934	64,960 ± 33,008	20,827 ± 0
Ammonia EC ^2^	5.61 ± 5.21	1.22 ± 0.59	5.95 ± 3.71	5.79 ± 0	11.37 ± 6.26	0.92 ± 0.48	7.19 ± 5.24	19.22 ± 0
Ammonia EF ^3^	0.381 ± 0.330	0.299 ± 0.090	0.288 ± 0.220	0.117 ± 0	2.974 ± 1.530	1.017 ± 0.540	2.203 ± 1.260	1.978 ± 0
TSP EC ^4^	379.35 ± 138.35	132.68 ± 78.44	472.73 ± 268.38	338.05 ± 0	735.03 ± 434.27	185.04 ± 191.56	268.91 ± 139.94	1814.36 ± 0
TSP EF ^3^	0.044 ± 0.041	0.048 ± 0.034	0.039 ± 0.025	0.009 ± 0	0.249 ± 0.093	0.282 ± 0.313	0.137 ± 0.125	0.247 ± 0
PM_10_ EC ^4^	279.71 ± 133.44	100.37 ± 81.63	229.60 ± 104.40	220.01 ± 0	474.67 ± 165.77	118.52 ± 101.62	187.64 ± 122.25	1131.87 ± 0
PM_10_ EF ^3^	0.029 ± 0.021	0.048 ± 0.052	0.018 ± 0.009	0.006 ± 0	0.133 ± 0.037	0.214 ± 0.237	0.077 ± 0.042	0.154 ± 0
PM_2.5_ EC ^4^	114.10 ± 41.87	52.45 ± 27.40	85.47 ± 39.30	60.53 ± 0	154.91 ± 98.53	113.63 ± 95.10	95.64 ± 34.70	383.34 ± 0
PM_2.5_ EF ^3^	0.011 ± 0.007	0.022 ± 0.014	0.007 ± 0.005	0.002 ± 0	0.054 ± 0.024	0.193 ± 0.144	0.045 ± 0.021	0.052 ± 0

^1^ VR: ventilation rate (m^3^ h^−1^); ^2^ EC: emission concentration (ppm); ^3^ EF: emission factor (kg animal^−1^ yr^−1^); ^4^ EC: emission concentration (μg m^−3^).

**Table 3 animals-13-03347-t003:** Correlation coefficients between ammonia emission factors and experimental variables (NA: number of animals, AA: age of animals in days, VR: ventilation rate in m^3^ h^−1^, OT: outdoor air temperature in °C, OH: outdoor relative humidity in %, IT: indoor air temperature in °C, IH: indoor relative humidity in %, OC: outdoor ammonia concentration in ppm, EC: emission concentration at the outlet in ppm, and EF: emission factor in kg animal^−1^ yr^−1^).

Variables	NA	AA	VR	OT	OH	IT	IH	OC	EC	EF
NA	1									
AA	0.929 ***	1								
VR	0.622 ***	0.680 ***	1							
OT	−0.039	0.033	0.488 ***	1						
OH	−0.007	0.075	0.143	0.167	1					
IT	−0.174	−0.034	0.498 ***	0.815 ***	0.294	1				
IH	−0.512 ***	−0.485 ***	−0.245	−0.217	0.290	0.054	1			
OC	−0.016	0.077	−0.006	0.189	0.143	0.077	−0.073	1		
EC	0.250	0.112	−0.340 *	−0.656 ***	−0.105	−0.674 ***	−0.085	−0.150	1	
EF	0.679 ***	0.613 ***	0.222	−0.115	−0.045	−0.248	−0.514 ***	−0.004	0.574 ***	1

* *p* < 0.05, *** *p* < 0.001.

**Table 4 animals-13-03347-t004:** Correlation coefficients between TSP emission factor and experimental variables (NA: number of animals, AA: age of animals in days, VR: ventilation rate in m^3^ h^−1^, OT: outdoor air temperature in °C, OH: outdoor relative humidity in %, IT: indoor air temperature in °C, IH: indoor relative humidity in %, OP: outdoor TSP concentration in μg m^−3^, OP_10_: outdoor PM_10_ concentration in μg m^−3^, OP_2.5_: outdoor PM_2.5_ concentration in μg m^−3^, EP: emission concentration at the outlet in μg m^−3^, and EF: emission factor in kg animal^−1^ yr^−1^).

Variables	NA	AA	VR	OT	OH	IT	IH	OP	OP_10_	OP_2.5_	EP	EF
OP	0.047	0.001	−0.140	−0.245	−0.383 *	−0.279	−0.152	1				
OP_10_	0.037	−0.025	−0.178	−0.305	−0.439 **	−0.326 *	−0.124	0.986 ***	1			
OP_2.5_	0.042	−0.048	−0.170	−0.317	−0.514 ***	−0.303	−0.117	0.657 ***	0.708 ***	1		
EP	0.262	0.175	−0.241	−0.635 ***	−0.297	−0.539 ***	−0.127	0.136	0.191	0.292	1	
EF	0.547 ***	0.581 ***	0.476 ***	0.070	−0.055	0.017	−0.529 ***	0.128	0.133	0.119	0.342 *	1

* *p* < 0.05, ** *p* < 0.01, *** *p* < 0.001.

**Table 5 animals-13-03347-t005:** Correlation coefficients between PM_10_ emission factor and experimental variables. (The abbreviations are the same as in Table 4).

Variables	NA	AA	VR	OT	OH	IT	IH	OP	OP_10_	OP_2.5_	EP	EF
EP	0.248	0.141	−0.252	−0.687 ***	−0.263	−0.503 ***	−0.049	0.241	0.297	0.340	1	
EF	0.488 ***	0.473 ***	0.638 ***	0.231	−0.067	0.322 *	−0.285	−0.050	−0.054	−0.003	0.213	1

* *p* < 0.05, *** *p* < 0.001.

**Table 6 animals-13-03347-t006:** Correlation coefficients between PM_2.5_ emission factor and experimental variables. (The abbreviations are the same as in Table 4).

Variables	NA	AA	VR	OT	OH	IT	IH	OP	OP10	OP2.5	EP	EF
EP	0.401 *	0.339 *	0.011	−0.468 **	−0.236	−0.411 **	−0.327 *	0.096	0.080	0.083	1	
EF	0.465 **	0.526 ***	0.735 ***	0.318 *	−0.034	0.313	−0.362 *	−0.084	−0.163	−0.121	0.395 *	1

* *p* < 0.05, ** *p* < 0.01, *** *p* < 0.001.

**Table 7 animals-13-03347-t007:** Correlation coefficients between emission concentrations and emission factors with MSY.

Substances	Spring	Summer	Autumn	Winter
Piglets EC ^1^	0.137	0.547 *	0.784 ***	0.297
Piglets EF ^2^	−0.044	0.037	0.346	−0.016
Growing–finishing pigs EC ^1^	0.502 *	0.451	0.380	−0.064
Growing–finishing pigs EF ^2^	0.462	−0.057	0.418	−0.311

^1^ EC: emission concentration; ^2^ EF: emission factor; * *p* < 0.05, *** *p* < 0.001.

## Data Availability

The data presented in this study are available on request from the corresponding author. The data are not publicly available due to privacy concerns.

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
