# Peer review of "Ammonia and Particulate Matter Emissions at a Korean Commercial Pig Farm and Influencing Factors"

_animals, 2023, doi:10.3390/ani13213347_

Round 1

Reviewer 1 Report

Comments and Suggestions for Authors

In the present work "Ammonia and particulate matters emissions in a Korean commercial pig farm and their influencing factors", Authors determinated emission factors of ammonia, TSP, PM10, and PM2.5 for piglets and growing-finishing pigs in a commercial pig farm in 31 Korea and identified the factors influencing the emission factors.

The topic is interesting, anyway some chenges have to be permormed in  order to go ongoing with the publication. 

The introduction is too short.  Please do a better overview of the topic and enrich the refereces according to the thematic.

I suggest you to do two part separated for results and discussion, if you want mantaining the division given in the results but discussing a part (in the section "discussion") what emerged and doing a comparison with the literature. 

The conclusions are too generics. Form line 502 to 519 you summarize the results obtained, this part could be more short. Instead, you have to enrich the part from line 528 to 532, giving concrete examples also reporting what is already available in the literature: more sustainable farming practices are already used in some contexts, to which type of specific pratictes are you referring to? Also about econimic part, you think that a lack of sustainable practices is only related with inadeguate contributes?

Author Response

Thank you for your valuable comments and feedback. Here are our responses to the questions and opinions you raised.

Reviewer 2 Report

Comments and Suggestions for Authors

Thank you for submitting your manuscript on Animal. Your paper takes piglets and fattening pigs as research objects to measure the emission factors of ammonia and particulate matter and their relationship with some indicators, which may provide a valuable reference for the sustainable construction of animal husbandry in the future. However, there are still some problems in the article. Please correct and solve the following problems carefully.

1.      Your research only focused on a single farm, why did you choose this farm as the research object? Did it have the significance of a contemporary representative farm?

2.      Add a space after methods in line 59. Add a space after facilities in line 71. Add a space after factor in line 281. Add a space after activity in line 283.

3.      Line 171-174, outdoor conditions of measurement, why choose an open hill? What is the distance from the farm? Whether there is a literature reference for the site selection?

4.      Line 185, check punctuation.

5.      Line 213, change where to Where

6.      Line 434, change PM10 and PM2.5 to PM10 and PM2.5.

7.      Line 478, May 2000 or May 2020?

8.      Line 589 and 623, check the format.

9.      As you mentioned in your Materials and Methods, only one test was conducted in winter, while 4-8 tests were conducted in other seasons. The data in winter appeared thin, so is it still convincing? 

10.   Why choose a period of one and a half years as the test period instead of a whole year?

11.   The grouping of animals in the Materials and Methods is not clear and detailed. Such as the number of animals, average age, etc.

12.   The first paragraph of the conclusion is a little redundant and should provide a concluding paragraph that clearly and concisely summarizes the results of this paper.

13.   The year should be bold in the references and the magazine name is not in the correct format. For name capitalization issues, such as line 650. Double-check the format of the reference list.

Comments on the Quality of English Language

The overall level is very high, but the expression of some sentences needs to be improved.

Author Response

(The authors gave the same response as above.)
